

**Effects of golf course management on subsurface soil properties in Iowa**
Matthew T. Streeter[*] and Keith E. Schilling
Iowa Geological Survey, 300 Trowbridge Hall, University of Iowa, Iowa City, IA 52242
*corresponding author: matthew-streeter@uiowa.edu; (319) 335-1593
**Abstract**
Currently, in the USA and especially in the Midwest region, urban expansion is developing
turfgrass landscapes surrounding commercial sites, homes, and recreational areas on soils that
have been agriculturally managed for decades. Often, golf courses are at the forefront of
conversations concerning anthropogenic environmental impacts since they account for some of
the most intensively managed soils in the world. Iowa golf courses provide an ideal location to
evaluate whether golf course management is affecting the quality of soils at depth. Our study
evaluated how soil properties relating to soil health and resiliency varied with depth at golf
courses across Iowa and interpreted relationships of these properties to current golf course
management, previous landuse, and inherent soil properties. Systematic variation in soil
properties including sand content, $NO_3$, and SOM were observed with depth at six Iowa golf
courses among three landform regions. Variability in sand content was identified between the
20 and 50 cm depth classes at all courses, where sand content decreased by as much as 37%.
Highest concentrations of SOM and $NO_3$ were found in the shallowest soils, whereas total C and
P variability was not related to golf course management. Sand content and $NO_3$ were found to
be directly related to golf course management, particularly at shallow depths. The effects of
golf course management dissipated with depth and deeper soil variations were primarily due to
natural geologic conditions.



## 1. Introduction


Critical evaluation of resilience and health of soil resources has been fueled by a recent
urgency to understand anthropogenic impacts on environmental resources relating to local,
regional, and international environmental sustainability and quality (Doran and Zeiss,
2000;Glanz, 1995). Urban expansion is a primary cause for these recent concerns. Currently, in
the USA and especially in the Midwest region, urban expansion is developing turfgrass
landscapes surrounding commercial sites, homes, and recreational areas on soils that have
been agriculturally managed for decades (Qian and Follett, 2002). Often, golf courses are at the
forefront of conversations concerning anthropogenic environmental impacts since they account
for some of the most intensively managed soils in the world (Balogh and Walker, 1992).
Limited information is available documenting changes in soils due to golf course
management, especially at depth, since most studies focus on the rooting depth of turfgrass
and near the surface where soils are engineered to ideal texture classes (Bauters et al., 2000).
Qian and Follett (2002) conducted a study on the effects of landuse change to turfgrass on soil
organic matter (SOM) and found that previous landuse imparted a strong baseline control.
However, while this study analyzed data from nearly 700 data sets, the results were limited to
the top 15 cm of soil. In contrast, deep soil quality affects via landuse management has been
observed in agricultural systems. Tomer and Burkart (2003) observed a significant increase in
soil nutrient concentrations at a depth of 17 m below ground surface that was associated with
historical fertilizer applications that occurred 20 years prior. Furthermore, they estimated $NO_3$-
N percolation rates in loess soils to be approximately 0.67 m/yr. With the apparent decades-
long translocation of soil nutrients after agricultural soil management, it is of interest to



determine whether golf course management is similarly affecting the resiliency and long-term
sustainability of soil resources.

There are over 400 golf courses in Iowa and many of these have been in operation for

decades. With some of the most productive agricultural soils in the world, Iowa golf courses
provide an ideal location to evaluate whether golf course management is affecting the quality
of soils at depth. Specifically, our study objectives were to: 1) evaluate how soil physical and
chemical properties relating to soil health and resiliency varied with depth at golf courses
across Iowa; and 2) interpret relationships of these properties to current golf course
management, previous landuse, and inherent soil properties.
**2. Materials and methods**

Our study sites consisted of 6 golf courses across Iowa, which were selected from

approximately 421 golf courses in the state. A stratified random design was used to select
courses by alphabetically grouping courses based on their location in eastern, central, and
western Iowa and separating them into 18-hole and 9-hole classes. Courses were then selected
using a random number generator. If permission for sampling was not obtained from the
course, the next randomly selected course was selected. Ultimately, three 18-hole and three 9-
hole courses were selected (Figure 1). Iowa is comprised of several landform regions that are
associated with different glacial advances and post-glacial erosion (Prior, 1991), and courses
were selected in a variety of landform regions. One course was located on the Iowan Surface
landform region in Eastern Iowa which consists of a significantly dissected glacial landscape and
loamy pedisediment. Two courses were located on the Southern Iowa Drift Plain, which is a
highly weathered pre-Illinoian glacial landscape capped in loess. Finally, three courses were



located on the Des Moines Lobe which consists of recent Wisconsinan age glacial deposits and
hummocky topography (Table 1).

Three of the courses (Central 9, West 9, and West 18) were combined public and private

use in towns of less than 5,000 people. The other three courses (East 9, East 18, and Central 18)
were privately managed country clubs positioned in communities of 10-60,000 people. The
smaller 9-hole courses ranged from 20 to 33 ha in size compared to the larger 18-hole courses
which ranged from 37 to 68 ha. Tee boxes, greens, and fairways at the courses consisted of
bentgrass except at the West 9 and Central 9 courses where fairways consisted of a variety of
bentgrass, bluegrass, and rye. Grass varieties in roughs varied widely but consisted mainly of
bentgrass, bluegrass, fescue, rye, and annual bluegrass in varying proportions. For each course,
superintendents recorded and made available annual fertilizer application rates (typically four
applications per year) for tee boxes, fairways, and roughs (Table 2). Generally, the roughs
received the least fertilizer while the tee boxes received the most. Four out of six roughs
received no fertilizer. Only one course (West 9) applied P fertilizer, whereas all courses applied
N and K fertilizer. Prior land use at the courses was identified from historical air photography
which was available starting in the 1930s (Iowa Department of Natural Resources, 2017). Golf
course opening dates were verified with the superintendents (Table 1).

At each course, soil samples were collected in conjunction with shallow monitoring well

installations being performed for a water quality study (Figure 2; Schilling and Streeter (2017)).
Based on conversation with course superintendents, sampling locations were selected that
could be easily accessed multiple times with minimal course disruption. Soil samples were
collected according to the stratigraphy encountered at each site. Sampling depths were



consistent within each course, but varied among courses based on the depth required to
breach the water table for the water quality assessment (Table 1).

Continuous core was collected from each borehole by which the soil was described

according to Schoeneberger (2012). Soil samples were collected, air dried and ground to pass a
2 mm sieve and lab analyzed according to Brown (1998) for P by strong Bray extraction.
Furthermore, total soil carbon (TC) and total soil nitrogen (TN) were determined by elemental
analysis via dry combustion and chromatography (Costech Analytical Technologies, 2015). Soil
organic Matter (SOM) was determined by weight loss on ignition (Walkley and Black, 1934) as
described by Schulte (1995). Nitrate nitrogen ($NO_3$) was measured by segmental flow analysis.
Particle size distribution was determined by x-ray absorption (Micromeritics Instrument
Corporation, 2015). A total of 127 soil samples were collected and analyzed (Table 1).
**3. Results**

Site stratigraphy was expectedly variable across the range of courses. The courses were

located on varying landscape positions and soils formed in a variety of different parent
materials including alluvium, glacial till, and loess. Depth of soil development ranged from 26
cm in glacial till at the central 18 course to 172 cm in loess at the west 18 course. Soils were less
variable within courses, but variations were observed due to golf course management,
primarily in terms of soil texture. In many locations, natural sand layers were identified at
varying depths as well as evidence of human alteration to soil texture near the soil surface. On
average, sand content was 25% higher in the upper 20 cm compared to the next lower depth
class (50 cm depth) for all courses (Table 3 and Figure 3). Decreasing sand content with depth
was most noticeable in courses on the Southern Iowa Drift Plain where average sand content



decreased from 45 to 8%. However, since background sand content in soils from different
geologic regions ranged widely among the courses (11-74%), we could not quantify the
statistical significance of the sand content differences due to the limited sample size.  In
essence, testing for changes in sand content in courses could be done within the same landform
region because soils have similar background conditions, but in our study we have only 1-3
courses per landform region.

In contrast to physical properties, varying concentrations of soil chemical properties

were identified near the soil surface at multiple courses (Table 3). Course average TC and SOM
ranged from 0.45-4.11% and 0.76-1.89%, respectively and ranged from 0.11-4.82% and 0.21-
5.50% for individual samples. Likewise, SOM ranged from 2.9 to 5.5% in samples collected at 20
cm, and was weakly correlated to TC (r=0.27). SOM decreased with depth at all courses (r=0.47)
regardless of landform region, whereas TC was not correlated to depth. Additional sampling is
required to test the statistical significance of changes in soil chemical properties with depth.
$NO_3$ and TN concentrations also decreased with increasing depth (r=0.41 and r=0.40,
respectively). $NO_3$ concentrations were observed to decrease at the greatest rates between the
first two depth classes (20 to 50 cm), whereas TN decreased more gradually. $NO_3$
concentrations decreased substantially from 15 to 2 mg/kg between 20 and 50 cm at East 18
located on the Iowan surface while concentrations dropped by more than 50% at all other
locations by 100 cm. C/N ratio ranged from 13 at the East 9 course (floodplain) to 57 at the
Central 18 course (glacial upland). C/N ratios of more than 20 generally occurred at depths
greater than 20 cm. However, C/N ratios were not significantly correlated with depth. Mean P



concentrations ranged from 18-67 mg/kg and were not correlated with any other soil property
measured.
**4. Discussion**
**4.1. Soil Physical Properties**
In this study, we evaluated soil conditions at six randomly selected Iowa golf courses
located in three different landform regions of the state.  Despite inherent variation associated
with parent materials, study results indicated systematic variation in several soil properties with
depth at the Iowa golf course sites. These systematic variations by depth are intriguing, but
require further study to expand sample sizes in order to test statistical significance. Even so,
preliminary conclusions may be drawn regarding the significance of landuse on soil properties.
At a gross scale, soil particle size distributions were mainly associated with the parent
material of the region. Typical sand contents of loess in western Iowa consist of less than 10%
sand, whereas soils formed in glacial till in central Iowa often exceed 50% sand and may be
highly variable (Soil Survey Staff, 2016). Our study found less than 10% sand in the unaltered
soil formed in loess from 20-100 cm at the West 18 course. However, the surface soil horizon
was much higher (45%) at this course. Unlike the deeper depth classes, the elevated sand
content near the soil surface was not likely an inherent soil property due to the nature of the
loess parent material, but rather caused by golf course landuse. Our study found sand content
was much higher at the four courses located on glacial till derived soils (West 9, Central 9 and
18, and East 18) compared to the courses located on loess (West 18) or alluvium (East 9) (Table
4). With the exception of the Central 9 course, the glacial derived soils exhibited a sharp
decrease in sand content between the 20 and 50 cm depths (as much as 37%) that may not be



explained primarily by inherent soil properties. Although natural variability in sand content
varied widely between landform regions, average sand content for all courses was 25% higher
in the 0-20 cm interval compared to the 20 to 50 cm depth.

Golf course soils are regularly aerated and top-dressed by adding new sand to the

surface of the established soil profile (Bandaranayake et al., 2003;Bauters et al., 2000). Golf
course superintendents at our study sites confirmed that topdressing was performed on an
annual basis at all but the Central 9 course. This topdressing management approach explains
the elevated sand content near the surface at our study sites. Increased sand content of the
surface soil horizons due to topdressing may have several implications for soil health and
development. Altering the particle size distribution of the soil may in-turn alter several soil
properties including bulk density and porosity (Arya and Paris, 1981;Gupta and Larson, 1979),
soil compaction (Bodman and Constantin, 1965), hydraulic conductivity and water holding
capacity (Campbell and Shiozawa, 1992;Jabro, 1992), as well as SOM content and distribution
(Anderson et al., 1981;Puget et al., 2000;Tiessen and Stewart, 1983), and $NO_3$ concentrations
via CEC (Anderson et al., 1981;Arya and Paris, 1981;Ersahin et al., 2006). Bandaranayake et al.
(2003) explained that topdressing soil with sand prolongs the time required for SOM content to
reach equilibrium in the soil profile. Since SOM affects several other soil properties, it is likely
that the soil conditions at our study sites are much less stable than those at surrounding non-
golf course sites.
**4.2. Soil chemical properties**

Similar to the inherent nature of soil texture, TC content may also be primarily

dependent on the soil parent material and may be highly spatially variable (Huang et al., 2007).



Courses on unaltered glacial deposits (West 9, Central 9 and 18) exceeded 4% TC due to high
inorganic carbon concentrations, whereas courses on reworked Iowan Surface glacial sediments
(East 18) as well as weathered Southern Iowa Drift Plain deposits had TC concentrations less
than 2%. These reworked and weathered soils have likely experienced inorganic carbon
leaching which has left the soils void of almost all inorganic carbon. This was quite noticeable
when comparing TC between landform regions at all depths and it also explains the variability
in C/N ratio at depths greater than 100 cm in our study where C/N ratio is 4 to 10 times greater
in Des Moines Lobe soils. The variability in TC (and C/N ratio below 100 cm) that we have
identified through this study may be entirely affected by parent material and natural
weathering patterns.

Soil chemical properties including SOM at the Iowa golf course were altered from their

long-term equilibrium conditions established under native perennial vegetation in response to
modern agricultural management and urban landuse. For example, agricultural drainage in
conjunction with row crop agriculture can decrease SOM between 24 and 89% compared to
that of perennially managed soils (Knops and Tilman, 2000;Kucharik et al., 2001;VandenBygaart
et al., 2003). In urban areas, Qian and Follett (2002) conducted a study on the effects of landuse
change to turfgrass and found that previous landuse imparted a strong baseline control on SOM
concentrations. Bandaranayake et al. (2003) estimated that turfgrass systems (e.g. golf courses)
could sequester up to 32 Mg/ha of soil organic carbon (approximately 64Mg/ha of SOM) in the
top 20 cm of soil within 30 years of establishment.  At the Iowa golf courses, we observed
substantially lower SOM contents in the upper 20 cm (2.9 to 5.5%) in samples collected at 20
cm but concentrations were highest in the uppermost layer and decreased with depth at all





courses.  Although accumulation of SOM in the upper 20 cm is consistent with Bandaranayake
et al. (2003), attributing specific SOM changes to golf course management is not possible with
our study. The SOM profiles could define native perennial, agricultural, or urban turfgrass
landuse.

In agricultural systems, application of N fertilizer has been found to migrate through the

soil profile (De Haan et al., 2017;Tomer and Burkart, 2003). In deep loess soils of western Iowa,
Tomer and Burkart (2003) documented a zone of high soil $NO_3$ located 17 m below land surface
due to over-application of commercial fertilizer 20 years earlier. De Haan et al. (2017) observed
that cropping systems has a large impact on residual soil $NO_3$ in the upper 1.8 m of the soil
profile. Residual soil $NO_3$ in the soil profile under continuous corn with a rye cover crop was
more than seven times higher than a perennial grass system. Others have similarly observed
residual $NO_3$ concentrations under agricultural land use in Iowa as high as 60 mg/kg (Blackmer
et al., 1989). However, the golf courses in our study applied varying rates of slow release N
fertilizer to Tee boxes and fairways, and only two courses (East 9 and 18) applied N fertilizer to
the roughs (Table 2). We estimated (based on typical bulk density values for surface soils in
Iowa) that soil $NO_3$ concentrations ranged from 15-35 kg/ha for our study sites. Our study
shows evidence of higher $NO_3$ levels in golf course soils than that of typical perennially
managed soils (less than 10 kg/ha), but not necessarily as high as typical agricultural
management (40-70 kg/ha) (De Haan et al., 2017). Recent studies provide evidence that once
application of N fertilizer ceases, the top 1 m of the soil may return to native concentrations of
$NO_3$ within 10 years (Streeter and Schilling, 2017). Since the courses for our study have been



established for over 50 years, it is likely that the current $NO_3$ concentrations near the soil
surface reflects golf course management rather than historical agricultural practices.

P concentrations at the courses were not correlated with any other soil property

measured during our study. Furthermore, P fertilizer was only applied at 1 course (West 9) on
the rough. Typically, background P concentrations in Iowa soils vary from 0-100 mg/kg
depending on the mineralogy of the soil parent material (Fenton, 1999). Loess derived soils
generally have the highest P, while glacial till derived soils will have some of the lowest P
concentrations (Fenton, 1999). Similarly, our study identified the highest P concentrations at
the West 18 course (loess derived soil) and the lowest P concentrations at the Central 18 course
(glacial till derived soil). Our study results suggest no significant alterations to background soil P
concentrations by golf course management.
**5. Conclusions**

In this study, systematic variation in soil properties including sand content, $NO_3$, and

SOM were observed with depth at six Iowa golf courses among three landform regions.
Variability in sand content was identified between the 20 and 50 cm depth classes at all
courses, where sand content decreased by as much as 37%. Highest concentrations of SOM and
$NO_3$ were found in the shallowest soils, whereas TC and P variability was not related to golf
course management and did not correlate with depth. Although many of the soil properties
measured for this study may be influenced by parent material and native vegetation, sand
content and $NO_3$ were found to be directly related to golf course management, particularly at
shallow depths. The effects of golf course management dissipated with depth and deeper soil
variations were primarily due to natural geologic conditions. Further work is recommended to




increase sample size of course sites within different landform regions to better quantify the
variability of soil properties with depth. Likewise, additional investigation of spatial patterns
within individual courses would improve characterization of soil quality patterns among
managed and unmanaged areas.  Despite limitations, our study indicates that golf course
management is affecting the surface and subsurface quality of soils.

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





**Figure Captions:**
Figure 1. Location map showing the six golf courses chosen for this study.
Figure 2. Soil sampling and monitoring well installation at East 18 course.
Figure 3. Profile of selected soil constituents with depth at the six Iowa golf courses.















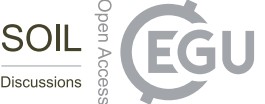

Table 1. Golf course site information.

| Course | Year Opened | Previous Landuse | Landform | Parent Material | Max. Depth (cm) | Number of Samples |
|--------|-------------|------------------|----------|-----------------|-----------------|-------------------|
| West 18 | 1963 | row crop | Southern Iowa Drift Plain | loess | 767 | 19 |
| West 9 | 1938 | row crop | Des Moines Lobe | glacial till | 380 | 20 |
| Central 9 | 1965 | row crop | Des Moines Lobe | glacial till | 457 | 18 |
| Central 18 | 1915 | no prior history | Des Moines Lobe | glacial till | 506 | 26 |
| East 18 | 1965 | row crop | Iowan Surface | glacial sediments | 546 | 22 |
| East 9 | 1920 | no prior history | Southern Iowa Drift Plain | alluvium | 405 | 22 |



















Table 2. Summary of annual fertilizer rates at selected Iowa golf courses.

| Course | Position | N kg/ha | Weighted Course Average N kg/ha | P kg/ha | K kg/ha |
|---|---|---|---|---|---|
| West 18 | Tee | 90 | | 0 | 44 |
| West 18 | Rough | 0 | | 0 | 0 |
| West 18 | Fairway | 77 | 28.5 | 0 | 11 |
| West 9 | Tee | 73 | | 5.4 | 22 |
| West 9 | Rough | 0 | | 0 | 0 |
| West 9 | Fairway | 37 | 15.4 | 0 | 6 |
| Central 18 | Tee | 146 | | 0 | 26 |
| Central 18 | Rough | 0 | | 0 | 0 |
| Central 18 | Fairway | 110 | 41.7 | 0 | 0 |
| Central 9 | Tee | 51 | | 0 | 17 |
| Central 9 | Rough | 0 | | 0 | 0 |
| Central 9 | Fairway | 51 | 18.5 | 0 | 17 |
| East 18 | Tee | 87 | | 0 | 132 |
| East 18 | Rough | 112 | | 0 | 17 |
| East 18 | Fairway | 63 | 95.8 | 0 | 107 |
| East 9 | Tee | 169 | | 0 | 20 |
| East 9 | Rough | 22 | | 0 | 7 |
| East 9 | Fairway | 115 | 59.2 | 0 | 19 |


















Table 3.  Soil conditions by depth at Iowa golf courses.

| Course | n | Depth Class | Sand % | P mg/kg | SOM % | NO3 mg/kg | TC % | TN % | C/N |
|---|---|---|---|---|---|---|---|---|---|
| Central 18 | 2 | 20 | 53 +/-42 | 18 +/-2 | 5.50 +/-4.24 | 5.00 +/-2.82 | 4.63 +/-0.17 | 0.33 +/-0.28 | 22 +/-19 |
| Central 18 | 5 | 50 | 40 +/-23 | 18 +/-9 | 4.46 +/-3.69 | 8.20 +/-6.22 | 4.82 +/-2.35 | 0.29 +/-0.21 | 47 +/-75 |
| Central 18 | 6 | 100 | 40 +/-21 | 03 +/-3 | 1.65 +/-1.83 | 3.83 +/-2.31 | 3.08 +/-2.18 | 0.10 +/-0.08 | 30 +/-33 |
| Central 18 | 12 | 500 | 26 +/-25 | 02 +/-2 | 0.48 +/-0.27 | 1.25 +/-0.45 | 4.21 +/-2.14 | 0.05 +/-0.05 | 84 +/-37 |
| Central 18 | 1 | 1000 | 05 +/- | 02 +/- | 0.30 +/- | 1.00 +/- | 4.69 +/- | 0.07 +/- | 67 +/- |
| Central 9 | 2 | 20 | 35 +/-2 | 14 +/-1 | 5.50 +/-0.14 | 15.50 +/- | 2.58 +/-0.48 | 0.18 +/-0.07 | 14 +/-4 |
| Central 9 | 2 | 50 | 50 +/-1 | 10 +/-0 | 2.35 +/-1.76 | 4.00 +/-4.24 | 1.81 +/-0.43 | 0.08 +/-0.01 | 22 +/-12 |
| Central 9 | 5 | 100 | 46 +/-10 | 04 +/-9 | 1.52 +/-1.13 | 1.60 +/-0.89 | 1.15 +/-0.84 | 0.06 +/-0.05 | 19 +/-1 |
| Central 9 | 9 | 500 | 61 +/-10 | 04 +/-3 | 0.48 +/-0.30 | 1.11 +/-0.33 | 1.51 +/-1.25 | 0.01 +/-0.02 | 15 +/-5 |
| East 18 | 1 | 20 | 81 +/- | 31 +/- | 2.60 +/- | 15.00 +/- | 1.54 +/- | 0.11 +/- | 14 +/- |
| East 18 | 6 | 50 | 74 +/-9 | 27 +/-1 | 1.53 +/-0.95 | 2.83 +/-2.31 | 0.77 +/-0.54 | 0.04 +/-0.03 | 19 +/-6 |
| East 18 | 4 | 100 | 78 +/-8 | 17 +/-1 | 0.40 +/-0.21 | 1.25 +/-0.50 | 0.18 +/-0.12 | 0.01 +/-0.00 | 18 +/-9 |
| East 18 | 9 | 500 | 79 +/-25 | 15 +/-3 | 0.21 +/-0.16 | 1.11 +/-0.33 | 0.11 +/-0.08 | | |
| East 18 | 2 | 1000 | 42 +/-0 | 02 +/-2 | 0.80 +/-0 | 1.00 +/-0 | 1.04 +/-0.10 | | |
| East 9 | 4 | 20 | 11 +/-5 | 16 +/-1 | 3.15 +/-1.96 | 4.50 +/-3.31 | 1.74 +/-1.77 | 0.10 +/-1.12 | 17 +/-22 |
| East 9 | 6 | 50 | 07 +/-2 | 23 +/-5 | 1.48 +/-0.42 | 1.00 +/-0 | 0.94 +/-0.36 | 0.09 +/-0.02 | 10 +/-2 |
| East 9 | 12 | 500 | 28 +/-33 | 27 +/-2 | 0.57 +/-0.26 | 1.00 +/-0 | 0.22 +/-0.08 | 0.02 +/-0.02 | 11 +/-1 |
| West 18 | 1 | 20 | 45 +/- | 97 +/- | 2.90 +/- | 9.00 +/- | 1.72 +/- | 0.14 +/- | 12 +/- |
| West 18 | 6 | 50 | 06 +/-11 | 28 +/-4 | 2.92 +/-0.95 | 5.83 +/-3.43 | 0.90 +/-0.75 | 0.04 +/-0.05 | 22 +/-22 |
| West 18 | 3 | 100 | 01 +/-0 | 46 +/-2 | 2.20 +/-0.85 | 3.67 +/-2.08 | 1.07 +/-0.58 | 0.08 +/-0.01 | 13 +/-5 |
| West 18 | 6 | 500 | 16 +/-36 | 98 +/-3 | 0.63 +/-0.35 | 1.17 +/-0.40 | 0.24 +/-0.12 | 0.01 +/-0.01 | 31 +/-36 |
| West 18 | 3 | 1000 | 10 +/-6 | 48 +/-3 | 0.67 +/-0.11 | 1.00 +/-0 | 0.20 +/-0.05 | 0.01 +/-0.01 | 20 +/- |
| West 9 | 1 | 20 | 26 +/- | 19 +/- | 2.90 +/- | 3.00 +/- | 3.12 +/- | 0.12 +/- | 26 +/- |
| West 9 | 3 | 50 | 17 +/-4 | 28 +/-9 | 4.97 +/-1.54 | 5.33 +/-2.08 | 3.04 +/-0.84 | 0.2 +/-0.07 | 15 +/-4 |
| West 9 | 5 | 100 | 33 +/-15 | 21 +/-2 | 1.56 +/-0.92 | 3.60 +/-2.61 | 2.27 +/-0.84 | 3.6 +/-0.05 | 23 +/-9 |
| West 9 | 11 | 500 | 33 +/-24 | 38 +/-4 | 0.76 +/-0.56 | 1.27 +/-0.64 | 1.87 +/-0.86 | 0.02 +/-0.03 | 19 +/-16 |















Table 4.  Mean soil conditions at Iowa golf courses.

| Course | Sand % | P mg/kg | SOM % | NO3 mg/kg | TC % | TN % | C/N |
|---|---|---|---|---|---|---|---|
| Central 18 | 33 +/-25 | 18 +/-25 | 1.89 +/-2.65 | 3.46 +/-3.87 | 4.11 +/-2.06 | 0.13 +/-0.15 | 57 +/-48 |
| Central 9 | 52 +/-12 | 50 +/-26 | 1.53 +/-1.74 | 3.16 +/-5.54 | 1.56 +/-1.05 | 0.04 +/-0.06 | 18 +/-6 |
| East 18 | 74 +/-19 | 37 +/-24 | 0.76 +/-0.84 | 2.22 +/-3.17 | 0.45 +/-0.50 | 0.01 +/-0.03 | 22 +/-7 |
| East 9 | 19 +/-26 | 57 +/-22 | 1.29 +/-1.26 | 1.63 +/-1.86 | 0.69 +/-0.91 | 0.05 +/-0.06 | 13 +/-13 |
| West 18 | 11 +/-22 | 67 +/-44 | 1.72 +/-1.24 | 3.42 +/-3.18 | 0.64 +/-0.63 | 0.03 +/-0.04 | 23 +/-22 |
| West 9 | 30 +/-20 | 31 +/-35 | 1.7 +/-1.69 | 2.55 +/-2.13 | 2.20 +/-0.91 | 0.06 +/-0.07 | 20 +/-10 |


























Figure 1.

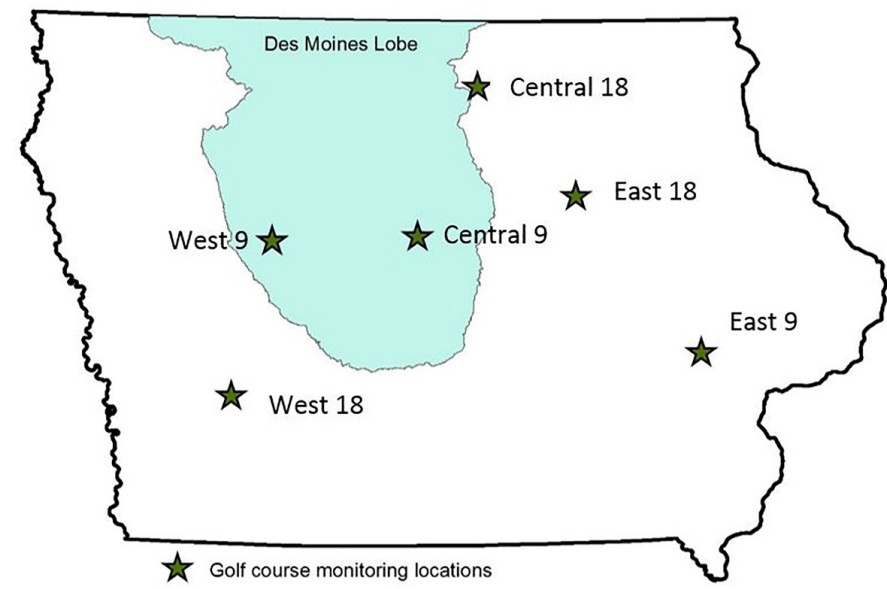





Figure 2.

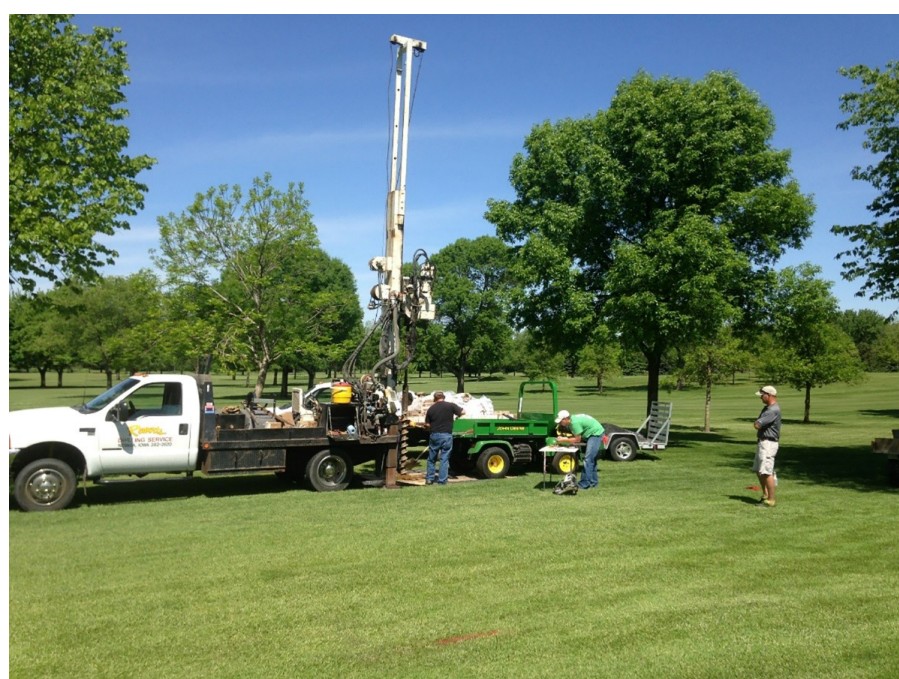





Figure 3.

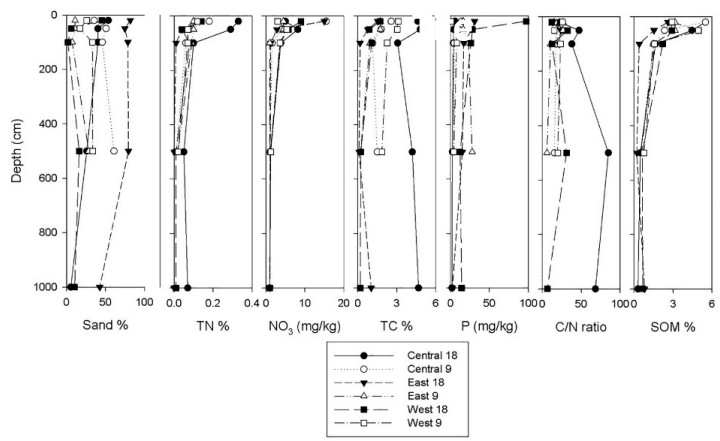


