# Peer review of "Effects of golf course management on subsurface soil properties in Iowa"

_SOIL, 2017_

## Referee Comment (RC1) · B. Prudat (Referee) · 3 Feb 2018

General comments: This original study deals with soil evolution under golf course. It tries to evaluate the anthropogenic impact on soils and to evaluate the management practices that influence sustainability of soil resource. To do so, the soils of various golf course with various soil management practices were analysed. The influence of management is highlighted using variation in depth. This method should be reconsidered, given that natural differences of soil properties with depth are high, probably higher than the differences that result from anthropogenic influence. The paper would be therefore highly improved by comparing, for each golf course (or courses in a similar landform), the differences between fairways, rough and tee (mostly between tee and fairways), which represent an intensification of soil use and management (e.g. fairways

not fertilised?). This could mostly be observed in the 0-20 cm layers, maybe in deeper layers under specific conditions. Line to line comments: L.10: Why is Iowa ideal? Briefly explain. L.97: Soil Organic Matter L.113-114: paired t-test would make it (compare 0-20cm to below) L.115-117: I don't think it is not possible to evaluate the textural changes due to golf management. I don't understand why you don't differentiate the fairways (natural?) to the other more intensive soil management. L.122 & 125 Need the p-value to support the correlation showed by the pearson r. L.122 The information is not very relevant as TC = inorgC+orgC. The correlation between orgC and TC should not be calculated without precaution and it doesn't bring much information apart that inorgC is not highly correlated to orgC... L.125 NO3 is highly mobile, therefore the last fertilisation date is often the most important information that influence NO3, at least in the 0-20cm. L.130 C/N should always express the ratio between organic C and total N, because this ratio was used to interpret the SOM "quality". Moreover, in depth, OC and N concentrations are often very low, too low to be accurately measured. Therefore, the value of C/N ratio is very inaccurate (especially if TN close to 0 mg/g). (take care in the interpretation of this value, especially L.183). L.140 Soil properties vary with depth. That is not intriguing at all, please rephrase to be more accurate, what is exactly intriguing? L156 In average 25% higher sand content. Risky to show the result that way because few samples with extreme values can strongly influence an average. Use statistical test to "prove" that sand content is higher (e.g. t-test). L180-181 Is this IC leaching a natural process or do you suggest that it is an influence of golf management? L193-195: Not clear, please rephrase. Higher... than? Table 3 Depth in mm, in cm? Please explain in the legend or make it clear in the table.

---

## Author Comment (AC1) · 6 Feb 2018

We would like to start by thanking the referee for the detailed comments and effort that has been put forth to review this manuscript. We were pleased that while there were several issues that needed addressed, these issues were mostly minor. In response to the general comments made which suggests a comparison by location on the golf course rather than comparisons between golf courses by depth: While it is true that management may vary between Tees, roughs, and fairways, we found through interviews with golf course managers that management varied much more significantly between courses than between locations on the course. For example, the two central courses were located on the same landform region, yet course average N fertilizer was more than doubled at Central 18 compared to Central 9. When looking at these same

two courses, N fertilization was the same for Tees and Fairways at the Central 9 course and was only 25% less on the fairway compared to the Tee on the Central 18 course (Table 2 in manuscript). Based on these types of observations, the decision was made to combine samples at golf courses and compare between courses where differences in management were most pronounced. Furthermore, as noted in the manuscript discussion starting at line 143, typical particle size content within parent materials in Iowa is well documented. In the case of loess derived soils in our western site, variability in particle size by depth may easily be attributed to anthropogenic factors (45% sand in the altered surface compared to less than 10% sand in the unaltered parent material). Likewise, with detailed soil description (which was done for each of our sites), soil alteration due to anthropogenic factors may be differentiated from natural pedogenesis visually (see attached image).

Response to specific comments, all grammatical suggestions have been noted and will be corrected in the revised manuscript. Line to line comments are as follows: Line 10 – Iowa is an ideal location because of the extent of "urban expansion that is developing turfgrass landscapes surrounding commercial sites, homes, and recreational areas on soils that have been agriculturally managed for decades" (lines6-8 in manuscript). Line 113-117 – In 5 of 6 of the courses, the 0-20 cm depth class had an n of 1 or 2. Unfortunately, this did not allow for statistical comparison by depth within individual courses by location (tee vs. fairway) and as stated in lines 112-117, the geologic differences between landform regions was too great to combine all surface samples. Line 122 and 125 – We will include p-values. Line 122 - The information related to TC is a relevant addition to this work because, In the case of our paper, we were not correlating organic and inorganic carbon (as the referee suggested), but correlating TC and SOM, which are derived via two separate testing procedures. This correlation helps to highlight the variability in parent material. Line 125 - Fertilization dates were recorded as well as soil sampling dates. This information may be added to the manuscript to clarify potential differences in nitrate due to sampling date. Line 130 - See comments for line 122 in justification of the comparison of total carbon to total nitrogen. We will acknowledge the

potential inaccuracy due to low nitrogen concentrations near line 183. Line 156 - See comments for line 113-117 regarding statistical power. Line 180-181 – "The variability in TC (and C/N ratio below 100 cm) that we have identified through this study may be entirely affected by parent material and natural weathering patterns" (lines 184-186).

———————————————————
[Figure]

[Figure]

**Fig. 1.**

---

## Referee Comment (RC2) · A. Baumgarten (Referee) · 22 Feb 2018

General comments: This short communication highlights the variability of certain soil characteristics at golf course sites, trying to attribute it either to anthropogenic activity or the natural soil development. Furthermore, the influence of the sustainability of the soil resource should be assessed. Firstly, most of the reported observations seem to be quite predictable, e.g. the higher amount of sand or available nitrogen in the upper layer. Secondly, the discussion and conclusion is only summing up the results without trying to deduce a possible ecological relevance of the data. Furthermore, I would also recommend a comparison more within the individual golf courses or between fairways, roughs and tees, maybe even using relations of results instead of absolute values.

[Figure]

Line to line comments L40: effects instead of affects? L 75ff: scientific names should be mentioned as well L95: TC is measured, but without determining the carbonate content. If this measurement would have been done, a further, rather important parameter would have been available and the Walkley Black – determination (which is, by the way, not state of the art any more) would have been unnecessary L98. If nitrate is determined from dried samples, the value does not correspond to the situation in situ any longer, as the drying process significantly influences the nitrate content L122: the weak correlation to TC is logical, as TC also comprises the anorganic carbon from carbonates. L150:ff Of course the golf course use promotes changes of sand content! But what are the implications? L158ff: This is why sand is added at golf courses! L183: For C/N-ratios, do not use TC but Corg, as the N is not linked to inorganic carbon at all! L 210: mg/kg soil? L367, Table 3: what does e.g. 53+/-42 mean (Sand % for Central 18, 20cm)? If it indicates a value of 53 with a variability of plus/minus 42, this is not very informative! Another example would be SOM at Central 18, 100cm depth: 1.65 +/-1.83. Try at least to adopt the layout or consider another way of displaying the data. For East9, the font size of the first two depth classes is not consistent with the table L 379, Table 4: see comments regarding layout of results for table 3

---

## Author Comment (AC2) · 22 Feb 2018

We would like to start by thanking the referee for the detailed comments and effort that has been put forth to review this manuscript. Similar to the initial reviewer's comments, most issues brought forth were minor. In response to the general comments made, which were similar to the first reviewer, which suggests a comparison by location on the golf course rather than comparisons between golf courses by depth: To reiterate from comments made in response to reviewer 1, while it is true that management may vary between tees, roughs, and fairways, we found through interviews with golf course managers that management varied much more significantly between courses than between locations on the course. For example, the two central courses were located on the same landform region, yet course average N fertilizer was more

than doubled at Central 18 compared to Central 9. When looking at these same two courses, N fertilization was the same for Tees and Fairways at the Central 9 course and was only 25% less on the fairway compared to the Tee on the Central 18 course (Table 2 in manuscript). Based on these types of observations, the decision was made to combine samples at golf courses and compare between courses where differences in management were most pronounced. Furthermore, as noted in the manuscript discussion starting at line 143, typical particle size content within parent materials in Iowa is well documented. In the case of loess derived soils in our western site, variability in particle size by depth may easily be attributed to anthropogenic factors (45% sand in the altered surface compared to less than 10% sand in the unaltered parent material). Likewise, with detailed soil description (which was done for each of our sites), soil alteration due to anthropogenic factors may be differentiated from natural pedogenesis visually (see attached image). Furthermore, while the reported observations may appear predictable in some cases, these data in spatial scale and depth are quite unique to Iowa and therefore, relevant and valuable to advancing the ecological sustainability of the region.

Response to specific comments, all grammatical suggestions have been noted and will be corrected in the revised manuscript. Line to line comments are as follows: Line 75 – scientific names will be added. Line 98 – While nitrate concentrations may have changed via the analytical process, in-situ concentrations were not specifically the goal, but rather, site to site comparisons. Line 150 – Implications of golf course management pertaining to addition of sand are explained in the paragraph starting at line 158, as self-noted by the reviewer. Lines 95, 122, 183 pertaining to TC - The information related to TC is a relevant addition to this work because, In the case of our paper, we were not correlating organic and inorganic carbon (as the first referee suggested), but correlating TC and SOM, which are derived via two separate testing procedures. This correlation helps to highlight the variability in parent material. "The variability in TC (and C/N ratio below 100 cm) that we have identified through this study may be entirely affected by parent material and natural weathering patterns" (lines 184-186). Since we

are making comparisons by depth in soils that are naturally calcareous, we chose to make our comparisons using TC, which helps to highlight these natural variations. Line 210 – Yes, mg/kg of soil. We will modify. Line 367, 379 – In the case of our study sites, natural variability is quite high. With limited sample size in a highly variable landscape, it is not uncommon to have large standard deviations (even greater than the mean in some cases). A note will be made in the description that +/- is referring to the standard deviation.

[Figure]

**Fig. 1.**

---

## Author Response (AR1)

**Response to Editor:**

Dear Editor,

First, thank you for your effort and detail in reviewing our manuscript. We have incorporated all of the line to line comments from the referees and have addressed all of the main concerns in this revised version of the manuscript. You will also find a response to your individual concerns below as well as in the submitted "mark-up" version of the manuscript found below.

1. Why the variation of soil properties in depth is consider in this research to be less than the expected variations due to anthropogenic impacts?

A: I must apologize that I do not fully understand the question. However, we have detailed the differences in the native soil properties compared to the ones that are impacted primarily by golf course management in the paragraph starting at line 164 in the manuscript. Our results show that human alteration is significantly greater than the natural variability.

2. Why variation within each location (golf course) is assumed to be less relevant than variations between locations? It is necessary to explain in the manuscript, particularly in the methods section all the evidences that the authors have in relation to this issue.

A: Thank you for bringing this important aspect forward. We have taken the opportunity to include an additional paragraph in the Methods section that details the decision to analyze between courses rather than within courses (lines 110-119 in the revised manuscript).

3. Please could you go deeper in the discussion on the possible ecological relevance of your results? As well as in the implications on the differences on sand content?

A: We have included information on ecological implications in each section of the discussion. These are mostly contained in lines: 195-201, 230-234, and 253-259.

4. Could you be a bit more conceptual and less descriptive in your conclusion? Which relevance for sustainability on medium and long term, for soil health has that the soils of golf courses are affected by management? You can introduce also this issue in the discussion.

A: We have added a conceptual portion to the conclusion that includes the relevance and implications of this preliminary research, supporting the need for a more detailed study. Similar changes have been included in the revised abstract.

[revised manuscript text omitted]